# Outcomes of Pregnancies and Deliveries of Patients Who Underwent Fertility-Preserving Surgery for Early-Stage Epithelial Ovarian Cancer

**DOI:** 10.3390/jcm11185346

**Published:** 2022-09-12

**Authors:** Shin Nishio, Takayo Takeno, Takeshi Fukuda, Ayumi Shikama, Hidekatsu Nakai, Hiroko Nakamura, Hideki Tokunaga, Kazuaki Takahashi, Emi Okuma, Masahiko Mori, Yasuhisa Terao, Kimio Ushijima, Nobuo Yaegashi

**Affiliations:** 1Department of Obstetrics and Gynecology, School of Medicine, Kurume University, Kurume 830-0011, Japan; 2Department of Obstetrics and Gynecology, Kagoshima City Hospital, Kagoshima 890-8760, Japan; 3Department of Obstetrics and Gynecology, School of Medicine, Osaka Metropolitan University, Osaka 545-8586, Japan; 4Department of Obstetrics and Gynecology, Faculty of Medicine, University of Tsukuba, Tsukuba 305-8576, Japan; 5Department of Obstetrics and Gynecology, Faculty of Medicine, Kindai Universit, Sayama 589-8511, Japan; 6Department of Obstetrics and Gynecology, National Hospital Organization, Kure Medical Center and Chugoku Cancer Center, Kure 737-0023, Japan; 7Department of Obstetrics and Gynecology, School of Medicine, Tohoku University, Sendai 980-8574, Japan; 8Department of Obstetrics and Gynecology, School of Medicine, The Jikei University, Tokyo 105-8471, Japan; 9Department of Obstetrics and Gynecology, Faculty of Medicine, Saga University, Saga 849-8501, Japan; 10Department of Gynecologic Oncology, Aichi Cancer Center Hospital, Nagoya 464-0021, Japan; 11Department of Obstetrics and Gynecology, Faculty of Medicine, Juntendo University, Tokyo 113-0033, Japan

**Keywords:** early-stage epithelial ovarian cancer, fertility-preserving surgery, perinatal prognosis, preterm birth

## Abstract

Some studies have shown increased risks of preterm birth, low birth weight, and cesarean delivery after oncologic treatment; others have shown the opposite. We evaluated the outcomes of pregnancies and deliveries of patients who underwent fertility-preserving surgery (FSS) for early-stage epithelial ovarian cancer (EOC) and examined their perinatal prognosis. This retrospective study included women with a history of stage IA or IC ovarian cancer reported in our previous study. The primary outcome was preterm birth after cancer diagnosis was considered. Secondary outcomes were neonatal morbidity and severe maternal morbidity. Thirty-one children were born to 25 women who had undergone FSS. The mean number of weeks at delivery was 38.7 ± 0.7, and the mean birth weight of infants was 3021 ± 160 g. With respect to pregnancy outcomes, 5 patients had preterm labor and 26 had full-term labor. The delivery mode was vaginal delivery in 18 patients and cesarean delivery in 13. Complications during pregnancy included placenta previa (one case) and pelvic abscess (one case). Except for three preterm infants with low birth weight, there were no other perinatal abnormalities. Pregnancy after fertility preservation in EOC has an excellent perinatal prognosis, although the cesarean delivery rate is high.

## 1. Introduction

The National Cancer Institute predicted that women younger than 45 years would account for approximately 12% of the predicted 21,410 new cases of ovarian cancer in 2021 in the United States [1]. As childbirth is increasingly delayed [2], the likelihood that a woman will be diagnosed with ovarian cancer and have a current or future desire to conceive also increases [3], and fertility-sparing surgery (FSS) is considered safe for carefully selected women with early-stage ovarian cancer [4,5]. Although a fertility-sparing treatment approach has a positive effect on quality of life [6,7], cancer survivors are less likely to conceive than their peers, even when their ability to do so is preserved [8,9], and this may be related to the fear of pregnancy and adverse obstetric outcomes [7].

Some population-based studies have shown increased risks of preterm birth [10,11,12,13,14], low birth weight [11,12,13,14], and cesarean delivery [12] after oncologic treatment; contrarily, others have demonstrated only some or none of these effects [15,16]. Moreover, it is unclear to what extent these studies can be extrapolated to patients with ovarian cancer because the majority of these studies have jointly analyzed all reproductive cancers [10,17] or included a few ovarian cancer patients [10,13], whereas other studies have not stratified patients by cancer type [12,14], or they have focused on survivors of childhood cancers [14,17].

In our previous study, the results suggested that fertility-preserving treatment may be safe for patients with stage IA epithelial ovarian cancer (EOC), clear cell carcinoma, and stage IC EOC, with or without adjuvant chemotherapy [18]. We are currently conducting a prospective non-randomized validation study to expand the indications for fertility-preserving treatment for EOC in JCOG 1203 for stage IA, clear cell, stage IC, and non-clear cell cancers [19]. In our previous study [18], we reported that 54 out of 84 (64.3%) patients who tried to conceive became pregnant, and 56 healthy children were born. However, no detailed studies exist on the perinatal period, including the background, means of conception, and pregnancy outcomes in pregnant cases. To evaluate fertility-preserving treatment, assessing both treatment and pregnancy outcomes is necessary. Therefore, the present study aimed to clarify the perinatal outcomes of patients who underwent FSS for EOC, in order to provide useful information for patients receiving this treatment.

## 2. Materials and Methods

This study was a preplanned secondary analysis of the dataset from a previous retrospective observational study performed between 1995 and 2007 [16]. The previous study was conducted by the Gynecologic Cancer Study Group of the Japan Clinical Oncology Group (JCOG-GCSG), and it was a multicenter retrospective observational study examining FSS for stage IA EOC, clear cell carcinoma, and stage IC EOC, with or without adjuvant chemotherapy, at 11 JCOG-GCSG-affiliated institutions. The study was hosted by the Kurume University School of Medicine (institutional review board approval registration number: 17229). Each participating center obtained ethical committee approval. The requirement for informed consent was waived, owing to the retrospective nature of the present study.

We collected the following data on pregnancy outcomes and perinatal outcomes: (1) age (at the beginning of pregnancy (years and months)); (2) height and weight at the time of pregnancy; (3) history of smoking, alcohol consumption, and oral contraceptive use; (4) marital status (never married, married, never remarried after divorce, never remarried after bereavement); (5) number of births, number of pregnancies; (6) age at menarche, menstrual cycle before treatment (regulated, irregular, unknown); (7) histological type (a. serous, b. mucinous, c. endometrioid, d. clear cell, e. others); (8) surgery type; (9) adjuvant chemotherapy (none, yes), chemotherapy regimen, number of cycles; (10) post-treatment menstrual cycle (restored to regularity, irregular but restored, not restored, unknown); (11) marital status after treatment (same as before treatment, married, divorced, bereaved, unknown); (12) time to pregnancy after treatment; (13) forms of pregnancy after treatment (natural pregnancy, methods of assisted reproductive technology (ART); (14) post-treatment period until the start date of ART; (15) date of return of pregnancy after treatment, duration of pregnancy (miscarriage, preterm labor, full-term labor); (16) perinatal events (presence or absence of cervical suture), delivery mode (vaginal delivery, cesarean delivery and indications), course of delivery, and presence or absence and type of uterine contraction inhibitors; (17) maternal complications during pregnancy after treatment (pregnancy-related complications (gestational hypertension, gestational diabetes, fetal failure to thrive, abnormal placental position, other maternal complications, and psychiatric disorders) and outcomes; (18) neonatal information (congenital diseases and other complications); and (19) date of last confirmed survival (disease-free survival (date of confirmed recurrence, with or without treatment of recurrence), treatment).

Comparison of continuous variables was performed using Student’s *t*-test. Fisher’s exact test was used for categorical variables, as appropriate, for each category size. Statistical significance was set at *p* < 0.05, unless otherwise stated. For survival analysis, data on progression-free survival (PFS) were censored from the date of surgery to the date of the last follow-up if disease progression had not occurred. An event was defined as death from any cause, disease relapse, or disease progression. Data on overall survival (OS) were censored from the date of surgery to the date of the last follow-up. An event was defined as death occurring from any cause. Oncologic outcomes, PFS, and OS were analyzed using the Kaplan–Meier method.

Statistical analysis was performed using SAS version 9.1 (SAS Institute Inc., Cary, NC, USA) and the revised version 2.7.0.

## 3. Results

We obtained information regarding age, EOC stage, histological characteristics of the tumor, treatment details, and follow-up period from 25 patients in the present study. In 25 patients with unilateral stage I EOC, the distribution of stages was as follows: stage IA, *n* = 17; stage IC1, *n* = 2; stage IC2, *n* = 3; and stage IC3, *n* = 3. Table 1 summarizes the main characteristics of patients and tumors. The mean patient age was 26.7 ± 5.9 years (range, 19–39 years). The median follow-up duration was 90 months (range, 18–160 months) from the initial FSS (Table 1).

All 25 patients underwent unilateral salpingo-oophorectomy. Surgical staging included careful inspection and palpation of peritoneal surfaces with biopsies of any suspected lesions and peritoneal washing cytology. No patients underwent endometrial curettage during surgery, although most patients underwent endometrial cytology or biopsy before surgery. If optimal surgical staging required at least omentectomy, in addition to unilateral salpingo-oophorectomy, all 25 patients were considered to be optimally staged.

Platinum-based adjuvant chemotherapy was administered to 18 (72%) patients, with a mean number of four cycles (range, three to six cycles). The most common chemotherapy regimens were cyclophosphamide + cisplatin (six of 18; 33.3%), cyclophosphamide + doxorubicin + cisplatin (five of 18; 27.8%), and paclitaxel + carboplatin (three of 18; 16.7%). The remaining four patients who received adjuvant chemotherapy were administered paclitaxel + carboplatin, cyclophosphamide + carboplatin, irinotecan + cisplatin, and fluorouracil, respectively. Seven (28%) patients received no adjuvant treatment after initial surgery (Table 1).

Recurrence was not identified in any patient during the follow-up period. The median follow-up duration for this group was 79 months. Twenty-five patients showed rates of 100% for 5-year PFS and OS. The median follow-up duration for these patients was 78 months.

In total, 31 children were born to the 25 women after surgery, with or without adjuvant chemotherapy, with a mean interval of 34 (8–48) months from cancer treatment to pregnancy. Five women (20%) in the total cohort who underwent FSS received ART treatment, according to medical records.

The mean maternal age at the time of delivery was 31.7 ± 2.1 years. All deliveries were singleton and occurred at full-term, at a mean gestational age of 38.7 ± 0.7 weeks. Eighteen (58.1%) of the vaginal deliveries were induced, and five of the planned cesarean deliveries were induced. Eight children were delivered via unplanned cesarean delivery. No congenital malformations were registered, and the mean birth weight was 3021 ± 160 g (Table 2). Table 3 and Table 4 provide details on pregnancy-related and fetal outcomes, respectively.

## 4. Discussion

Early-stage EOC is a relatively uncommon disease in young women; hence, this study adds to the current body of knowledge by reporting on both the safety and efficacy of FSS. The 100% PFS and OS rates at 5 years were in accordance with previously published data [20,21,22].

Compared to perinatal reports from Japan [23], in this study, women who conceived after FSS for stage IA or IC ovarian cancer did not have an increased risk of preterm birth, delivery of small-for-gestational-age (SGA) neonates, neonatal morbidity, or severe maternal morbidity; however, the rates of cesarean delivery were higher. In this study, the cesarean delivery rate was 42%, which is clearly higher than the rate of 18.5% in the general population [23]. In this study, the majority of cesarean deliveries were due to delivery arrest; nevertheless, the apparent reason for this was unclear.

Receipt of chemotherapy did not appreciably affect the proportion of adverse obstetric events. US guidelines have highlighted the importance of discussing fertility preservation with young cancer patients for at least a decade [24,25], but data regarding obstetric outcomes after FSS have been limited. Our study provides encouraging evidence that pregnancy after FSS in stage IA or IC ovarian cancer is generally safe.

Given the rarity of ovarian cancer, even studies that focused on patients with reproductive cancers who conceived included small numbers of ovarian cancer patients [10,26]. A systematic review of obstetric outcomes after reproductive cancers demonstrated that most studies were a case series with few births, and more than one-third of studies did not comment on the viability or gestational age at birth [26]. In another study, women with a history of reproductive cancer had a greater absolute risk of preterm birth than women in a matched control group, but this difference may have been driven by the 28% of cervical cancer survivors who delivered prematurely [10]. Furthermore, we included only patients who conceived after their treatment, unlike prior reports, which included patients who were diagnosed with cancer during pregnancy [11], a group that is likely at higher risk of adverse obstetric outcomes, iatrogenic or otherwise. Considering that the preterm birth rate in this study was 16.1% (five patients at 34–36 weeks), our results may be more applicable to women who are contemplating pregnancy after completion of ovarian cancer treatment.

Our results vary from prior data that suggested a possible increase in neonatal complications after treatment for early-stage ovarian cancer [14].

Data that guide the timing of pregnancy after cancer treatment are sparse. It has been suggested that cancer patients—particularly those receiving chemotherapy—postpone conception until 12–24 months after treatment completion [13], given the possible damage to oocytes and prolonged immunosuppression, which could predispose patients to preterm birth, SGA neonates, and miscarriage [27]. In the current study, we did not find that chemotherapy recipients, or those who delivered within a year of diagnosis, had higher frequencies of adverse events, although these analyses were limited by sample size. Hence, these important questions should be investigated in future studies with longer follow-up periods.

This study had several limitations. First, the sample size was small; thus, the results must be interpreted with caution. Moreover, only half of the originally planned number of patients could be analyzed, due to the deliveries at another facility and lack of follow-up data. Second, several patients from more than 20 years ago were included, and there may be discrepancies with current treatment. Finally, the true impact of FSS on pregnancies is not known because only cases that resulted in live births were included, and there are no data on miscarriages.

In conclusion, our study results provide important insights to guide shared decision-making discussions regarding FSS for patients with early-stage ovarian cancer. These data may reassure patients considering FSS that pregnancy after ovarian cancer treatment is not associated with increased rates of preterm birth and neonatal morbidity, although the risk of cesarean delivery is higher.

## Figures and Tables

**Table 1 jcm-11-05346-t001:** Patient characteristics and oncologic outcomes (*n* = 25).

Factor	*n* (%)
**Age,** **mean (range), years**	26.7 ± 5.9 (19–39)
**FIGO stage (2014)**	
IA	17 (68)
IC1	2 (8)
IC2	3 (12)
IC3	3 (12)
**Histology**	
Mucinous	16 (64)
Serous	7 (28)
Clear	2 (8)
**Surgery**	
Unilateral salpingo-oophorectomy	25
Contralateral ovary biopsy	13
Omentectomy	13
Pelvic lymph node biopsy	4
Pelvic lymphadenectomy	1
Pelvic lymphadenectomy + para-aortic lymphadenectomy	3
No pelvic lymph node retrieval	17
**Adjuvant therapy**	
None	7 (28)
Yes (chemotherapy)	18 (72)
Regimen	
Cyclophosphamide + cisplatin	6
Cyclophosphamide + doxorubicin + cisplatin	5
Paclitaxel + carboplatin	3
Weekly paclitaxel + carboplatin	1
Cyclophosphamide + carboplatin	1
Irinotecan + cisplatin	1
Fluorouracil	1

Abbreviation: FIGO, International Federation of Gynecology and Obstetrics.

**Table 2 jcm-11-05346-t002:** Obstetrical outcomes of 25 women who gave birth (31 children) after FSS.

Factor	*n* (%)
**Age at the time of pregnancy, mean, years**	31.7 ± 2.1
**Pregnancy method, *n* (%)**	
Spontaneous	26 (73.9)
Assisted reproductive technology	5 (16.1)
**Delivery mode, *n* (%)**	
Vaginal	18 (58.1)
Planned cesarean delivery	5 (16.1)
Unplanned cesarean delivery	8 (25.8)
**Births, *n***	
Single	29
Twins	2
**Gestational age at birth (weeks), mean**	38.7 ± 0.6
**Child weight at birth (g), mean**	3021 ± 160
**Sex of child, *n* (%)**	
Male	21 (67.7)
Female	10 (32.3)
**Delivery outcomes, *n* (%)**	
Preterm	5 (16.1)
Term	26 (83.9)
**Complications during pregnancy, *n* (%)**	2 (6.5)
Placenta previa, *n*	1
Pelvic abscess, *n*	1
**Complications among children, *n* (%)**	4 (12.9)
Low birth weight infant, *n*	4

**Table 3 jcm-11-05346-t003:** Pregnancy-related outcomes.

Number	Maternal Age at the Time of Pregnancy	Time from Surgery to Pregnancy	Pregnancy Method	Gestational Age at Birth (Weeks)	Delivery Mode	Indication for C/D	Complications during Pregnancy
1	37y6mo	2y4mo	ART	39w4d	Vaginal		None
2	33y4mo	13y1mo	ART	40w4d	C/D	Delivery arrest	None
^a^ 3	29y8mo	9y8mo	Spontaneous	37w2d	Vaginal		None
^a^ 4	31y6mo		Spontaneous	35w0d	C/D	Infection	Pelvic abscess
5	29y6mo	8y7mo	Spontaneous	39w2d	Vaginal		None
6	40y7mo	2y3mo	Spontaneous	38w2d	Vaginal		Threatened miscarriage
7	32y0mo	3y5mo	Spontaneous	38w6d	Vaginal		None
^b^ 8	25y3mo	2y0mo	Spontaneous	39w6d	C/D	Delivery arrest	None
^b^ 9	29y1mo		Spontaneous	38w3d	C/D	Previous C/D	None
^c^ 10	33y10mo	1y4mo	Spontaneous	40w5d	C/D	Delivery arrest	None
^c^ 11	37y11mo		Spontaneous	38w3d	C/D	Previous C/D	None
^d^ 12	37y2mo	5y7mo	ART	39w6d	Vaginal		Threatened miscarriage
^d^ 13	40y3mo		ART	40w3d	Vaginal		Vasa previa
14	22y6mo	2y9mo	Spontaneous	39w5d	Vaginal		None
15	27y8mo	2y8mo	Spontaneous	38w6d	Vaginal		None
16	36y2mo	6y0mo	Spontaneous	38w4d	Vaginal		None
17	28y3mo	5y2mo	Spontaneous	39w4d	Vaginal		None
18	39y3mo	1y6mo	Spontaneous	39w3d	C/D	Fetal distress	None
^e^ 19	30y0mo	5y5mo	Spontaneous	36w1d	C/D	DD twin	Threatened labor
^e^ 20	30y0mo		Spontaneous	36w1d	C/D	DD twin	Threatened labor
^f^ 21	24y2mo	1y2mo	Spontaneous	39w3d	Vaginal		None
^f^ 22	29y11mo		Spontaneous	39w6d	Vaginal		None
23	40y4mo	1y4mo	Spontaneous	37w5d	Vaginal		None
24	31y6mo	3y10mo	Spontaneous	39w6d	Vaginal		None
25	33y5mo	3y7mo	Spontaneous	36w6d	Vaginal		None
26	28y11mo	2y2mo	Spontaneous	39w1d	C/D	Delivery arrest	Threatened miscarriage
27	34y4mo	5y3mo	Spontaneous	39w6d	Vaginal		None
28	23y4mo	1y7mo	Spontaneous	39w3d	Vaginal		None
29	24y9mo	5y6mo	Spontaneous	38w0d	C/D	Delivery arrest	None
30	23y10mo	1y9mo	Spontaneous	40w5d	C/D	CPD	None
31	40y11mo	1y8mo	ART	34w1d	C/D	Placenta previa	Threatened labor/placenta previa

Abbreviations: y, years; mo, months; ART, assisted reproductive technology; C/D, cesarean delivery; DD, dichorionic diamniotic, ^a, b, c, d, e,^ and ^f^ are identical patients.

**Table 4 jcm-11-05346-t004:** Fetal outcomes.

Number	Child Weight at Birth (g)	Sex	Delivery Mode	Complications of Children
1	2952	Male	Vaginal	None
2	2828	Female	C/D	None
^a^ 3	3022	Male	Vaginal	None
^a^ 4	2226	Male	C/D	Low birth weight infant
5	2710	Male	Vaginal	None
6	3162	Male	Vaginal	None
7	3674	Female	Vaginal	None
^b^ 8	3716	Female	C/D	None
^b^ 9	2818	Male	C/D	None
^c^ 10	3561	Male	C/D	None
^c^ 11	2810	Male	C/D	None
^d^ 12	3375	Male	Vaginal	None
^d^ 13	3785	Male	Vaginal	None
14	3109	Male	Vaginal	None
15	3216	Male	Vaginal	None
16	3429	Female	Vaginal	None
17	2977	Female	Vaginal	None
18	3362	Male	C/D	None
^e^ 19	2448	Female	C/D	None
^e^ 20	2044	Female	C/D	None
^f^ 21	2877	Male	Vaginal	None
^f^ 22	2675	Female	Vaginal	None
23	2920	Male	Vaginal	None
24	2795	Female	Vaginal	None
25	2905	Male	Vaginal	None
26	3286	Male	C/D	None
27	3320	Male	Vaginal	None
28	2947	Male	Vaginal	None
29	2698	Male	C/D	None
30	3586	Male	C/D	None
31	2418	Female	C/D	Low birth weight infant

^a, b, c, d, e^ and ^f^ are identical patients. Abbreviation: C/D, cesarean delivery.

## Data Availability

Not applicable.

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
