# Peer review of "Outcomes of Pregnancies and Deliveries of Patients Who Underwent Fertility-Preserving Surgery for Early-Stage Epithelial Ovarian Cancer"

_jcm, 2022, doi:10.3390/jcm11185346_

Round 1

Reviewer 1 Report

This is paper reports on an interesting and important topic.  However, sample size is small and therefore the findings must be treated with caution. 

Results - it would be helpful in either Table 1 or the text to report the median age as well as the mean. 

The numbers are unclear - in the first 4 paragraphs the study includes 25 patients but then in Table 2 and associated paragraphs there are 31 patients who had FSS.  How do these 2 numbers relate?  Are the 25 a subset of the 31 or in addition to?  This could be clarified with a flow chart.

The sentence that reads" A total of 31 children were born to nine women who had given birth" is unclear.  Does this mean that 9 women gave birth to the 31 children or 9 women had had a delivery prior to the FSS?

The paper does not comment on the completeness of follow-up.  Could the women have had pregnances/deliveries at another hospital and so would have been missed in the medical record review?

It may be useful to include some details in Table 3 about the type of cancer treatment and surgery that the woman received.  I think you also need to include time to pregnancy - given that this is commented on in the discussion but data is not presented to back this up.

Author Response

Dear Dr. Andrès, 

Thank you for giving us the opportunity to submit a revised draft of our manuscript titled “Outcomes of pregnancies and deliveries of patients who underwent fertility-preserving surgery for early-stage epithelial ovarian cancer to the Journal of Clinical Medicine. We appreciate the time and effort that you and the reviewers have dedicated to providing your valuable feedback on my manuscript and are grateful to the reviewers for their insightful comments on our paper. We have been able to incorporate changes to reflect most of the suggestions provided by the reviewers. Point-by-point response to the reviewers’ comments and concerns are given below.

Reviewer 1

This is paper reports on an interesting and important topic.  However, sample size is small and therefore the findings must be treated with caution. 

Response: Thank you for your appreciation. We acknowledge that the sample size is small and have mentioned that the results must be interpreted with caution in the limitations paragraph.

Results - it would be helpful in either Table 1 or the text to report the median age as well as the mean. 

Response: Thank you for pointing this out. We have specified the mean age in Line 118 and Table 1.

The numbers are unclear - in the first 4 paragraphs the study includes 25 patients but then in Table 2 and associated paragraphs there are 31 patients who had FSS.  How do these 2 numbers relate?  Are the 25 a subset of the 31 or in addition to?  This could be clarified with a flow chart.

Response: We apologize for the confusion and lack of clarity. In our study, there were 25 patients with 31 deliveries. We have performed our calculations again and revised the manuscript accordingly.

The sentence that reads" A total of 31 children were born to nine women who had given birth" is unclear.  Does this mean that 9 women gave birth to the 31 children or 9 women had had a delivery prior to the FSS?

Response: We apologize for the confusion and lack of clarity. In our study, there were 25 patients with 31 deliveries. We have now clarified this in the revised manuscript.

The paper does not comment on the completeness of follow-up.  Could the women have had pregnances/deliveries at another hospital and so would have been missed in the medical record review?

Response: Thank you for pointing this out. In our study, only half of the originally planned number of patients could be analyzed because of deliveries at another facility and lack of follow-up data. We have clarified this in the limitations paragraph (Lines 200-202).

It may be useful to include some details in Table 3 about the type of cancer treatment and surgery that the woman received.  I think you also need to include time to pregnancy - given that this is commented on in the discussion but data is not presented to back this up.

Response: Thank you for the recommendation. We have added details about the types of surgeries in Table 1 and the time from surgery to pregnancy in Table 3. With regard to the type of cancer, all women had early-stage epithelial ovarian cancer, which is already mentioned in the text.

Reviewer 2 Report

Summary of the Study

In this manuscript, Nishio et al. gathered pregnancy data from patients with ovarian cancer and found that there is no association between cancer treatment with preterm birth or other birth defects. Although the manuscript is well written, the authors need to explain/change a few things to improve it.

Comments to Authors:

  • Line 51: Add citations to studies that have not shown any effects on the pregnancies of cancer survivors.
  • In the method section, the authors described different statistical methods (t-test/Fisher’s test/survival analysis) that have been used but none of the results are shown or described in the paper.
  • Authors should make some figures to better compare the results (boxplots, etc.).

Author Response

Dear Dr. Andrès, 

Thank you for giving us the opportunity to submit a revised draft of our manuscript titled “Outcomes of pregnancies and deliveries of patients who underwent fertility-preserving surgery for early-stage epithelial ovarian cancer to the Journal of Clinical Medicine. We appreciate the time and effort that you and the reviewers have dedicated to providing your valuable feedback on my manuscript and are grateful to the reviewers for their insightful comments on our paper. We have been able to incorporate changes to reflect most of the suggestions provided by the reviewers. Point-by-point response to the reviewers’ comments and concerns are given below.

Reviewer 2

In this manuscript, Nishio et al. gathered pregnancy data from patients with ovarian cancer and found that there is no association between cancer treatment with preterm birth or other birth defects. Although the manuscript is well written, the authors need to explain/change a few things to improve it.

Response: Thank you for the positive comments regarding the language. We have attempted to address all your concerns in the revised version of the manuscript.

Comments to Authors:

Line 51: Add citations to studies that have not shown any effects on the pregnancies of cancer survivors.

Response: Thank you for the helpful suggestion. We have added references 15 and 16 in the revised manuscript (Line 52).

In the method section, the authors described different statistical methods (t-test/Fisher’s test/survival analysis) that have been used but none of the results are shown or described in the paper.

Authors should make some figures to better compare the results (boxplots, etc.).

Response: We apologize for the confusion. We initially planned to conduct the statistical analyses; however, because of the small sample size, the analyses had to be shelved halfway through the research. We have deleted the paragraphs on statistical analyses from the revised manuscript.

Round 2

Reviewer 2 Report

I thank the author for updating the manuscript. The updated manuscript is looking good, and the authors have addressed all of my concerns.